# META-LEARNING WITH IMPLICIT PROCESSES

## ABSTRACT

This paper presents a novel *implicit process-based meta-learning* (IPML) algorithm that, in contrast to existing works, explicitly represents each task as a continuous latent vector and models its probabilistic belief within the highly expressive IP framework. Unfortunately, meta-training in IPML is computationally challenging due to its need to perform intractable exact IP inference in task adaptation. To resolve this, we propose a novel expectation-maximization algorithm based on the stochastic gradient Hamiltonian Monte Carlo sampling method to perform meta-training. Our delicate design of the neural network architecture for meta-training in IPML allows competitive meta-learning performance to be achieved. Unlike existing works, IPML offers the benefits of being amenable to the characterization of a principled distance measure between tasks using the maximum mean discrepancy, active task selection without needing the assumption of known task contexts, and synthetic task generation by modeling task-dependent input distributions. Empirical evaluation on benchmark datasets shows that IPML outperforms existing Bayesian meta-learning algorithms. We have also empirically demonstrated on an e-commerce company's real-world dataset that IPML outperforms the baselines and identifies "outlier" tasks which can potentially degrade meta-testing performance.

## 1 INTRODUCTION

Few-shot learning (also known as *meta-learning*) is a defining characteristic of human intelligence. Its goal is to leverage the experiences from previous tasks to form a model (represented by meta-parameters) that can rapidly adapt to a new task using only a limited quantity of its training data. A number of meta-learning algorithms (Finn et al., 2018; Jerfel et al., 2019; Ravi & Beatson, 2018; Rusu et al., 2019; Yoon et al., 2018) have recently adopted a probabilistic perspective to characterize the uncertainty in the predictions via a Bayesian treatment of the meta-parameters. Though they can consequently represent different tasks with different values of meta-parameters, it is not clear how or whether they are naturally amenable to (a) the characterization of a principled similarity/distance measure between tasks (e.g., for identifying outlier tasks that can potentially hurt training for the new task, procuring the most valuable/similar tasks/datasets to the new task, detecting task distribution shift, among others), (b) active task selection given a limited budget of expensive task queries (see Appendix A.2.3 for an example of a real-world use case), and (c) synthetic task/dataset generation in privacy-aware applications without revealing the real data or for augmenting a limited number of previous tasks to improve generalization performance.

To tackle the above challenge, this paper presents a novel *implicit process-based meta-learning* (IPML) algorithm (Sec. 3) that, in contrast to existing works, explicitly represents each task as a continuous latent vector and models its probabilistic belief within the highly expressive IP[1] framework (Sec. 2). Unfortunately, meta-training in IPML is computationally challenging due to its need to perform intractable exact IP inference in task adaptation.[2] To resolve this, we propose a novel

---

[1] An IP (Ma et al., 2019) is a stochastic process such that every finite collection of random variables has an implicitly defined joint prior distribution. Some typical examples of IP include Gaussian processes, Bayesian neural networks, neural processes (Garnelo et al., 2018), among others. An IP is formally defined in Def. 1.

[2] The work of Ma et al. (2019) uses the well-studied Gaussian process as the variational family to perform variational inference in general applications of IP, which sacrifices the flexibility and expressivity of IP by constraining the distributions of the function outputs to be Gaussian. Such a straightforward application of IP to meta-learning has not yielded satisfactory results in our experiments (see Appendix A.4).

*expectation-maximization* (EM) algorithm to perform meta-training (Sec. 3.1): In the E step, we perform task adaptation using the stochastic gradient Hamiltonian Monte Carlo sampling method (Chen et al., 2014) to draw samples from IP posterior beliefs for all meta-training tasks, which eliminates the need to learn a latent encoder (Garnelo et al., 2018). In the M step, we optimize the meta-learning objective w.r.t. the meta-parameters using these samples. Our delicate design of the neural network architecture for meta-training in IPML allows competitive meta-learning performance to be achieved (Sec. 3.2). Our IPML algorithm offers the benefits of being amenable to (a) the characterization of a principled distance measure between tasks using maximum mean discrepancy (Gretton et al., 2012), (b) active task selection without needing the assumption of known task contexts in (Kaddour et al., 2020), and (c) synthetic task generation by modeling task-dependent input distributions (Sec. 3.3).

## 2 BACKGROUND AND NOTATIONS

For simplicity, the inputs (outputs) for all tasks are assumed to belong to the same input (output) space. Consider meta-learning on probabilistic regression tasks:[3] Each task is generated from a task distribution and associated with a dataset $(\mathcal{X}, \mathbf{y}_{\mathcal{X}})$ where the set $\mathcal{X}$ and the vector $\mathbf{y}_{\mathcal{X}} \triangleq (y_{\mathbf{x}})_{\mathbf{x} \in \mathcal{X}}^{\top}$ denote, respectively, the input vectors and the corresponding noisy outputs

$$y_{\mathbf{x}} \triangleq f(\mathbf{x}) + \epsilon(\mathbf{x}) \tag{1}$$

which are outputs of an unknown underlying function $f$ corrupted by an i.i.d. Gaussian noise $\epsilon(\mathbf{x}) \sim \mathcal{N}(0, \sigma^2)$ with variance $\sigma^2$. Let $f$ be distributed by an *implicit process* (IP), as follows:

**Definition 1** (Implicit process for meta-learning). *Let the collection of random variables $f(\cdot)$ denote an IP parameterized by meta-parameters $\theta$, that is, every finite collection $\{f(\mathbf{x})\}_{\mathbf{x} \in \mathcal{X}}$ has a joint prior distribution $p(\mathbf{f}_{\mathcal{X}} \triangleq (f(\mathbf{x}))_{\mathbf{x} \in \mathcal{X}}^{\top})$ implicitly defined by the following generative model:*

$$\mathbf{z} \sim p(\mathbf{z}), \quad f(\mathbf{x}) \triangleq g_{\theta}(\mathbf{x}, \mathbf{z}) \tag{2}$$

*for all $\mathbf{x} \in \mathcal{X}$ where $\mathbf{z}$ is a latent task vector to be explained below and generator $g_{\theta}$ can be an arbitrary model (e.g., deep neural network) parameterized by meta-parameters $\theta$.*

Definition 1 defines valid stochastic processes if $\mathbf{z}$ is finite dimensional (Ma et al., 2019). Though, in reality, a task may follow an unknown distribution, we assume the existence of an unknown function that maps each task to a latent task vector $\mathbf{z}$ satisfying the desired known distribution $p(\mathbf{z})$, like in (Kaddour et al., 2020).[4] Using $p(\mathbf{y}_{\mathcal{X}}|\mathbf{f}_{\mathcal{X}}) = \mathcal{N}(\mathbf{f}_{\mathcal{X}}, \sigma^2 \mathbf{I})$ (1) and the IP prior belief $p(\mathbf{f}_{\mathcal{X}})$ from Def. 1, we can derive the marginal likelihood $p(\mathbf{y}_{\mathcal{X}})$ by marginalizing out $\mathbf{f}_{\mathcal{X}}$.

**Remark 1.** Two sources of uncertainty exist in $p(\mathbf{y}_{\mathcal{X}})$: *Aleatoric uncertainty* in $p(\mathbf{y}_{\mathcal{X}}|\mathbf{f}_{\mathcal{X}})$ reflects the noise (i.e., modeled in (1)) inherent in the dataset, while *epistemic uncertainty* in the IP prior belief $p(\mathbf{f}_{\mathcal{X}})$ reflects the model uncertainty arising from the *latent task prior belief* $p(\mathbf{z})$ in (2).[5]

Let the sets $\mathcal{T}$ and $\mathcal{T}_*$ denote the meta-training and meta-testing tasks, respectively. Following the convention in (Finn et al., 2018; Gordon et al., 2019; Ravi & Beatson, 2018; Yoon et al., 2018), for each meta-training task $t \in \mathcal{T}$, we consider a support-query (or train-test) split of its dataset $(\mathcal{X}_t, \mathbf{y}_{\mathcal{X}_t})$ into the *support set* (or training dataset) $(\mathcal{X}_t^s, \mathbf{y}_{\mathcal{X}_t^s})$ and *query set* (or test/evaluation dataset) $(\mathcal{X}_t^q, \mathbf{y}_{\mathcal{X}_t^q})$ where $\mathcal{X}_t = \mathcal{X}_t^s \cup \mathcal{X}_t^q$ and $\mathcal{X}_t^s \cap \mathcal{X}_t^q = \emptyset$. Specifically, for a $N$-way $K$-shot classification problem, the support set has $K$ examples per class and $N$ classes in total.

Meta-learning can be defined as an optimization problem (Finn et al., 2017; 2018) and its goal is to learn meta-parameters $\theta$ that maximize the following objective defined over all meta-training tasks:

$$\mathcal{J}_{\text{meta}} \triangleq \log \prod_{t \in \mathcal{T}} p(\mathbf{y}_{\mathcal{X}_t^q}|\mathbf{y}_{\mathcal{X}_t^s}) = \sum_{t \in \mathcal{T}} \log \int p(\mathbf{y}_{\mathcal{X}_t^q}|\mathbf{f}_{\mathcal{X}_t^q}) \, p(\mathbf{f}_{\mathcal{X}_t^q}|\mathbf{y}_{\mathcal{X}_t^s}) \, \mathrm{d}\mathbf{f}_{\mathcal{X}_t^q} . \tag{3}$$

Task *adaptation* $p(\mathbf{f}_{\mathcal{X}_t^q}|\mathbf{y}_{\mathcal{X}_t^s})$ is performed via IP inference after observing the support set:

$$p(\mathbf{f}_{\mathcal{X}_t^q}|\mathbf{y}_{\mathcal{X}_t^s}) = \int_{\mathbf{z}} p(\mathbf{f}_{\mathcal{X}_t^q}|\mathbf{z}) \, p(\mathbf{z}|\mathbf{y}_{\mathcal{X}_t^s}) \, \mathrm{d}\mathbf{z} . \tag{4}$$

---

[3]We defer the discussion of meta-learning on probabilistic classification tasks using the robust-max likelihood (Hernández-Lobato et al., 2011) to Appendix A.1.

[4]$p(\mathbf{z})$ is often assumed to be a simple distribution like multivariate Gaussian $\mathcal{N}(\mathbf{0}, \mathbf{I})$ (Garnelo et al., 2018).

[5]Our work here considers a point estimate of meta-parameters $\theta$ instead of a Bayesian treatment of $\theta$ (Finn et al., 2018; Yoon et al., 2018). This allows us to interpret the epistemic uncertainty in $p(\mathbf{f}_{\mathcal{X}})$ via $p(\mathbf{z})$ directly.

The objective $\mathcal{J}_{\text{meta}}$ (3) is the "test" likelihood on the query set, which reflects the idea of "learning to learn" by assessing the effectiveness of "learning on the support set" through the query set. An alternative interpretation views $p(\mathbf{f}_{\mathcal{X}_t^q}|\mathbf{y}_{\mathcal{X}_t^s})$ as an "informative prior" after observing the support set. The objective $\mathcal{J}_{\text{meta}}$ (3) is also known as the Bayesian held-out likelihood (Gordon et al., 2019). In a meta-testing task, adaptation is also performed via IP inference after observing its support set and evaluated on its query set. Similar to GP or any stochastic process, the input vectors of the dataset are assumed to be known/fixed beforehand. We will relax this assumption by allowing them to be unknown when our IPML algorithm is exploited for synthetic task generation (Sec. 3.3).

## 3 IMPLICIT PROCESS-BASED META-LEARNING (IPML)

### 3.1 EXPECTATION MAXIMIZATION (EM) ALGORITHM FOR IPML

Recall that task adaptation requires evaluating $p(\mathbf{f}_{\mathcal{X}_t^q}|\mathbf{y}_{\mathcal{X}_t^s})$ (4). From Def. 1, if generator $g_\theta$ (2) can be an arbitrary model (e.g., deep neural network), then $p(\mathbf{f}_{\mathcal{X}_t^q}|\mathbf{y}_{\mathcal{X}_t^s})$ and $p(\mathbf{f}_{\mathcal{X}_t^q})$ cannot be evaluated in closed form and have to be approximated by samples. Inspired by the Monte Carlo EM algorithm (Wei & Tanner, 1990) which utilizes posterior samples to obtain a maximum likelihood estimate of some hyperparameters, we propose an EM algorithm for IPML: The E step uses the *stochastic gradient Hamiltonian Monte Carlo* (SGHMC) sampling method to draw samples from $p(\mathbf{f}_{\mathcal{X}_t^q}|\mathbf{y}_{\mathcal{X}_t^s})$ (4), while the M step maximizes the meta-learning objective $\mathcal{J}_{\text{meta}}$ (3) w.r.t. meta-parameters $\theta$:

**Expectation (E) step.** Note that since $\mathbf{f}_{\mathcal{X}_t^q} = (g_\theta(\mathbf{x}, \mathbf{z}))_{\mathbf{x} \in \mathcal{X}_t^q}^\top$ (2), no uncertainty exists in $p(\mathbf{f}_{\mathcal{X}_t^q}|\mathbf{z})$ in (4). So, $p(\mathbf{f}_{\mathcal{X}_t^q}|\mathbf{y}_{\mathcal{X}_t^s})$ can be evaluated using the same generator $g_\theta$ (2) and the *latent task posterior belief* $p(\mathbf{z}|\mathbf{y}_{\mathcal{X}_t^s})$, as follows:

**Remark 2.** Drawing samples from $p(\mathbf{f}_{\mathcal{X}_t^q}|\mathbf{y}_{\mathcal{X}_t^s})$ is thus equivalent to first drawing samples of $\mathbf{z}$ from $p(\mathbf{z}|\mathbf{y}_{\mathcal{X}_t^s})$ and then passing them as inputs to generator $g_\theta$ to obtain samples of $\mathbf{f}_{\mathcal{X}_t^q}$. Hence, given a task $t$, adaptation $p(\mathbf{f}_{\mathcal{X}_t^q}|\mathbf{y}_{\mathcal{X}_t^s})$ (4) essentially reduces to a task identification problem by performing IP inference to obtain the latent task posterior belief $p(\mathbf{z}|\mathbf{y}_{\mathcal{X}_t^s})$. This is a direct consequence of epistemic uncertainty arising from $p(\mathbf{z}|\mathbf{y}_{\mathcal{X}_t^s})$ and $p(\mathbf{z})$ (Remark 1).

In general, $p(\mathbf{z}|\mathbf{y}_{\mathcal{X}_t^s})$ also cannot be evaluated in closed form. Instead of using *variational inference* (VI) and approximating $p(\mathbf{z}|\mathbf{y}_{\mathcal{X}_t^s})$ with a potentially restrictive variational distribution (Garnelo et al., 2018; Kaddour et al., 2020; Ma et al., 2019), we draw samples from $p(\mathbf{z}|\mathbf{y}_{\mathcal{X}_t^s})$ using SGHMC (Chen et al., 2014). SGHMC introduces an auxiliary random vector $\mathbf{r}$ and samples from a joint distribution $p(\mathbf{z}, \mathbf{r}|\mathbf{y}_{\mathcal{X}_t^s})$ following the Hamiltonian dynamics (Brooks et al., 2011; Neal, 1993): $p(\mathbf{z}, \mathbf{r}|\mathbf{y}_{\mathcal{X}_t^s}) \propto \exp(-U(\mathbf{z}) - 0.5\mathbf{r}^\top \mathbf{M}^{-1}\mathbf{r})$ where the negative log-probability $U(\mathbf{z}) \triangleq -\log p(\mathbf{z}|\mathbf{y}_{\mathcal{X}_t^s})$ resembles the potential energy and $\mathbf{r}$ resembles the momentum. SGHMC updates $\mathbf{z}$ and $\mathbf{r}$, as follows:

$$\Delta\mathbf{z} = \alpha\mathbf{M}^{-1}\mathbf{r}, \quad \Delta\mathbf{r} = -\alpha\nabla_{\mathbf{z}}U(\mathbf{z}) - \alpha\mathbf{C}\mathbf{M}^{-1}\mathbf{r} + \mathcal{N}(\mathbf{0}, 2\alpha(\mathbf{C} - \mathbf{B}))$$

where $\alpha$, $\mathbf{C}$, $\mathbf{M}$, and $\mathbf{B}$ are the step size, friction term, mass matrix, and Fisher information matrix, respectively.[6] Note that $\nabla_{\mathbf{z}}U(\mathbf{z}) = -\nabla_{\mathbf{z}}\log p(\mathbf{z}|\mathbf{y}_{\mathcal{X}_t^s}) = -\nabla_{\mathbf{z}}\log p(\mathbf{z}, \mathbf{y}_{\mathcal{X}_t^s}) = -\nabla_{\mathbf{z}}[\log p(\mathbf{y}_{\mathcal{X}_t^s}|\mathbf{f}_{\mathcal{X}_t^s} = (g_\theta(\mathbf{x}, \mathbf{z}))_{\mathbf{x} \in \mathcal{X}_t^s}^\top) + \log p(\mathbf{z})]$ can be evaluated tractably.

**Maximization (M) step.** We optimize $\mathcal{J}_{\text{meta}}$ (3) w.r.t. $\theta$ using samples of $\mathbf{z}$. The original objective $\mathcal{J}_{\text{meta}} = \sum_{t \in \mathcal{T}} \log(\mathbb{E}_{p(\mathbf{z}|\mathbf{y}_{\mathcal{X}_t^s})}[p(\mathbf{y}_{\mathcal{X}_t^q}|\mathbf{f}_{\mathcal{X}_t^q} = (g_\theta(\mathbf{x}, \mathbf{z}))_{\mathbf{x} \in \mathcal{X}_t^q}^\top)])$ is not amenable to stochastic optimization with data minibatches, which is usually not an issue in a few-shot learning setting. When a huge number of data points and samples of $\mathbf{z}$ are considered, we can resort to optimizing the lower bound $\mathcal{J}_{\text{s-meta}}$ of $\mathcal{J}_{\text{meta}}$ by applying the Jensen's inequality:

$$\mathcal{J}_{\text{meta}} \geq \mathcal{J}_{\text{s-meta}} \triangleq \sum_{t \in \mathcal{T}} \mathbb{E}_{p(\mathbf{f}_{\mathcal{X}_t^q}|\mathbf{y}_{\mathcal{X}_t^s})}\big[\log p(\mathbf{y}_{\mathcal{X}_t^q}|\mathbf{f}_{\mathcal{X}_t^q})\big] = \sum_{t \in \mathcal{T}} \mathbb{E}_{p(\mathbf{z}|\mathbf{y}_{\mathcal{X}_t^s})}\big[\log p(\mathbf{y}_{\mathcal{X}_t^q}|\mathbf{f}_{\mathcal{X}_t^q})\big].$$

---

[6]The sampler hyperparameters $\alpha$, $\mathbf{C}$, $\mathbf{M}$, and $\mathbf{B}$ are set according to the auto-tuning method of Springenberg et al. (2016) which has been verified to work well in our experiments; more details are given in Appendix A.2.1.

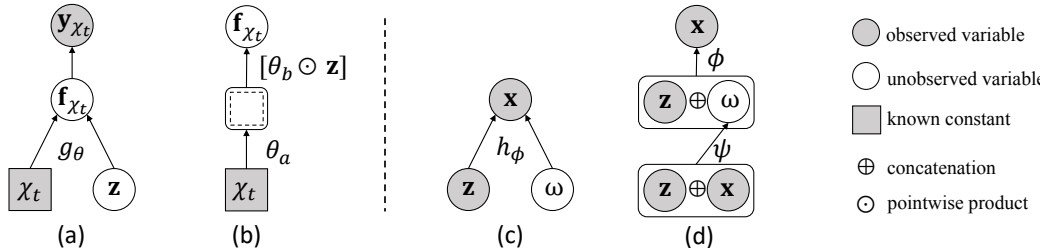

Figure 1: (a) Graphical model corresponding to IPML. (b) DNN implementation of generator $g_\theta$ where $\theta \triangleq (\theta_a, \theta_b)$ and $\theta_a$ can be convolutions to obtain high-level representations of the input vector, while $\theta_b$ is the last DNN layer's parameters which are masked by $\mathbf{z}$ during the forward passes. (c) Graphical model corresponding to input generation by X-Net. (d) CVAE implementation of X-Net (i.e., decoder neural network with parameters $\phi$).

## 3.2 ARCHITECTURE DESIGN FOR META-TRAINING

Our generator $g_\theta$ is implemented using a *deep neural network* (DNN) parameterized by meta-parameters $\theta$. Under this setup, we have empirically observed that the design of the coupling of $\mathbf{z}$ with the DNN $g_\theta(\mathbf{x}, \cdot)$ is crucial to achieving competitive performance of our IPML algorithm. A naive design by concatenating $\mathbf{z}$ with $\mathbf{x}$ (or higher-level abstractions of $\mathbf{x}$) as a contextual input during forward passes has not worked well as the resulting gradients w.r.t. $\mathbf{z}$ may not have provided enough guidance for SGHMC to learn a sufficiently useful representation of $\mathbf{z}$ in meta-training.

To this end, inspired by the *attention* mechanism (Vaswani et al., 2017) and *dropout* method (Srivastava et al., 2014), we introduce a design of the coupling by applying $\mathbf{z}$ as a *mask* to the last DNN layer's parameters: The last DNN layer's parameters are first masked by $\mathbf{z}$ (i.e., point-wise product with $\mathbf{z}$), as illustrated in Figs. 1a and 1b. Different tasks can now be distinguished by different masks, hence resembling different attentions on the last DNN layer's connections during forward propagation. We adopt soft masks[7] (i.e., continuous values) instead of hard masks (i.e., either 0 or 1). Such a design of the coupling is empirically demonstrated to be effective in our experiments (Appendix A.4.3).

## 3.3 ARCHITECTURE DESIGN FOR SYNTHETIC TASK GENERATION

Recall the assumption of known/fixed input vectors in $\mathcal{X}_t$ in the last paragraph of Sec. 2[8] which we will have to relax here. Synthetic task generation can be performed by the following procedure if $\mathbf{x}$ is task-independent (e.g., $p(\mathbf{x}, \mathbf{z}) = p(\mathbf{x})p(\mathbf{z})$): After meta-training is completed (Sec. 2), draw a sample of latent task vector $\mathbf{z} \sim p(\mathbf{z})$, draw samples of $\mathbf{x} \sim p(\mathbf{x})$ to form $\mathcal{X}_t$, and then generate noisy outputs $\mathbf{y}_{\mathcal{X}_t} = (g_\theta(\mathbf{x}, \mathbf{z}) + \epsilon(\mathbf{x}))^\top_{\mathbf{x} \in \mathcal{X}_t}$ to obtain the dataset $(\mathcal{X}_t, \mathbf{y}_{\mathcal{X}_t})$ for task $t$.

When $\mathbf{x}$ is task-dependent (e.g., for image classifications of different objects, $p(\mathbf{x}, \mathbf{z}) \neq p(\mathbf{x})p(\mathbf{z})$), not modeling $p(\mathbf{x}|\mathbf{z})$ limits the ability to generate $t$-dependent $\mathcal{X}_t$. To resolve this, our IPML algorithm includes an *X-generative network* (X-Net): $\mathbf{x} \triangleq h_\phi(\mathbf{z}, \boldsymbol{\omega})$ that learns to generate an input vector $\mathbf{x}$ given samples of the latent task vector $\mathbf{z}$ and random vector $\boldsymbol{\omega} \sim p(\boldsymbol{\omega}) = \mathcal{N}(\mathbf{0}, \mathbf{I})$ where $\boldsymbol{\omega}$ models the diversity of the input distribution given a fixed task represented by the sample of $\mathbf{z}$. There are several options to implement X-Net: Note that during the training of X-Net, both $\mathcal{X}_t$ and the samples of $\mathbf{z} \sim p(\mathbf{z}|\mathbf{y}_{\mathcal{X}_t^s})$ for all meta-training task $t \in \mathcal{T}$ are available. So, generative models such as the *conditional variational autoencoder* (CVAE) (Sohn et al., 2015) or conditional generative adversarial networks (Mirza & Osindero, 2014) are suitable for X-Net as they can utilize $\mathbf{z}$ as the *contextual* information. Our work here uses (the decoder of) CVAE to implement X-Net. Figs. 1c and 1d illustrate such a design. We have empirically observed that a simple concatenation with $\mathbf{z}$ suffices here as our delicate architecture design for meta-training (Sec. 3.2) can yield a useful representation of $\mathbf{z}$ for training X-Net well. Further details and a method to ensure balanced data generation are given in Appendix A.5. The training objective for synthetic task generation is the

---

[7]The latent task prior belief $p(\mathbf{z})$ is thus assumed to be a multivariate Gaussian $\mathcal{N}(\mathbf{1}, \mathbf{I})$.

[8]This assumption is reasonable for meta-training since only $p(\mathbf{y}_\mathcal{X})$ (and not $p(\mathbf{x})$) needs to be modeled.

empirical lower bound (Sohn et al., 2015) of VI on $p(\boldsymbol{\omega}|\mathbf{x}, \mathbf{z})$:

$$\mathcal{J}_{\mathrm{X}} \triangleq \sum_{t \in \mathcal{T}} \mathbb{E}_{\mathbf{z} \sim p(\mathbf{z}|\mathbf{y}_{\mathcal{X}_t^s})} \left[ |\mathcal{X}_t|^{-1} \sum_{\mathbf{x} \in \mathcal{X}_t} \left( \mathbb{E}_{q_\psi(\boldsymbol{\omega}|\mathbf{x}, \mathbf{z})}[\log p_\phi(\mathbf{x}|\mathbf{z}, \boldsymbol{\omega})] - D_{\mathrm{KL}}[q_\psi(\boldsymbol{\omega}|\mathbf{x}, \mathbf{z}) \| p(\boldsymbol{\omega})] \right) \right]$$

where $\phi$ and $\psi$ are, respectively, the parameters of X-Net (decoder neural network) and the encoder neural network, and $D_{\mathrm{KL}}$ denotes the KL distance. In the training of X-Net, we sample one $\mathbf{z}$ per update. We also sample one $\boldsymbol{\omega}$ per update to train with reparameterization tricks. Algorithms 1 and 2 describe meta-training (with training of X-Net) and synthetic task generation, respectively.

---

**Algorithm 1:** IPML: Meta-Training

**while** *not converged* **do**
  Sample task $t$ from $\mathcal{T}$
  E step: Sample $\{\mathbf{z}_1, \dots, \mathbf{z}_n\}$ with SGHMC
  M step: Sample $\mathbf{z}$ from $\{\mathbf{z}_1, \dots, \mathbf{z}_n\}$
        $\theta \leftarrow \theta + \eta \nabla_\theta \mathcal{J}_{\mathrm{meta}}$
  Update X-Net with $\mathbf{z}$ and $\mathcal{X}_t$:
      $\phi \leftarrow \phi + \eta \nabla_\phi \mathcal{J}_{\mathrm{X}}$, $\quad \psi \leftarrow \psi + \eta \nabla_\psi \mathcal{J}_{\mathrm{X}}$
**return** $\theta, \phi, \psi$

---

**Algorithm 2:** Synthetic Task Generation

Sample $\mathbf{z} \sim p(\mathbf{z})$
Initialize synthetic task $t$ and $\mathcal{X}_t = \emptyset$
**for** $i = 1, \dots,$ final size of $\mathcal{X}_t$ **do**
    Sample $\boldsymbol{\omega} \sim \mathcal{N}(\mathbf{0}, \mathbf{I})$
    Compute $\mathbf{x} = h_\phi(\mathbf{z}, \boldsymbol{\omega})$
    Compute $y_{\mathbf{x}} = g_\theta(\mathbf{x}, \mathbf{z}) + \epsilon(\mathbf{x})$
    $(\mathcal{X}_t, \mathbf{y}_{\mathcal{X}_t}) \leftarrow (\mathcal{X}_t \cup \{\mathbf{x}\}, \mathbf{y}_{\mathcal{X}_t \cup \{\mathbf{x}\}})$
**return** $(\mathcal{X}_t, \mathbf{y}_{\mathcal{X}_t})$ for task $t$

---

## 4 EXPERIMENTS AND DISCUSSION

**Benchmark datasets: sinusoid regression and few-shot image classification.** We first empirically evaluate the performance of our IPML algorithm against that of several Bayesian meta-learning baselines like the *neural process* (NP) (Garnelo et al., 2018), *Bayesian model-agnostic meta-learning* (BMAML) (Yoon et al., 2018), PLATIPUS (Finn et al., 2018), and *amortized Bayesian meta-learning* (ABML) (Ravi & Beatson, 2018) on benchmark meta-learning datasets. For few-shot image classification, we also empirically compare IPML with a strong baseline: *prototypical network* (PN) (Snell et al., 2017). We run experiments on three datasets: sinusoid, Omniglot (Lake et al., 2011), and mini-ImageNet (Ravi & Larochelle, 2017). Sinusoid is a regression task of sine waves with uniformly sampled amplitude in $[0.1, 5.0]$, phase in $[0, \pi]$, and input $\mathbf{x}$ in $[-5, 5]$. The generator of IPML and the baseline regressors are neural networks with 2 hidden layers of size 40 with ReLU nonlinearities. The Omniglot dataset consists of 20 instances of 1623 characters from 50 different alphabets. The mini-ImageNet dataset involves 64 training classes, 12 validation classes, and 24 test classes. For Omniglot and mini-ImageNet, our implementation and baselines all use the same data pre-processing, same train-test split, and same data augmentation as that in (Finn et al., 2017). The generator of IPML and the baseline classifiers are convolutional neural networks with 4 modules of $3 \times 3$ convolutions and 64 filters, followed by batch normalization, ReLU nonlinearities, and strided convolutions (Omniglot) or $2 \times 2$ max-pooling (mini-ImageNet). More details of the experimental settings can be found in Appendix A.2.2.

For sinusoid regression (Table 1), IPML outperforms MAML and BMAML by a fair margin. For Omniglot (Table 2), IPML is competitive with MAML and PN. For mini-ImageNet (Table 3), IPML outperforms MAML and all tested Bayesian meta-learning algorithms,[9] while being competitive with PN. PN achieves a higher classification accuracy for 1-shot 20-way Omniglot and 5-shot 5-way mini-ImageNet because PN utilizes more information from the extra classes during training (Snell et al., 2017). Specifically, though meta-testing involves $N$-way classification for all tested algorithms, the training of PN requires more than $N$ classes, that is, 60-way classification which is also the setting adopted in (Snell et al., 2017). As a result, since PN utilizes more information from the extra classes during training, it is reasonable to expect that PN achieves a higher classification accuracy at times. Overall, IPML is effective for benchmark datasets.

For both sinusoid regression (Table 1) and Omniglot (Table 2), NP performs unsatisfactorily as compared to IPML, likely because (a) it performs amortized variational inference of $\mathbf{z}$ through a heavily parameterized encoder which may introduce optimization difficulties and overfitting during meta-training, and (b) the encoder of NP takes in the simple concatenation of $(\mathbf{x}, y_{\mathbf{x}})$ and thus does not explicitly capture the $\mathbf{x} \rightarrow y_{\mathbf{x}}$ relationship in the support set.[10]

---

[9]Some of the results are taken from (Finn et al., 2018; Nguyen et al., 2020; Yoon et al., 2018). The 5-shot 5-way results for PLATIPUS and ABML are missing because there are no publicly available implementations.

[10]An ablation study of the limitations of NP can be found in Appendix A.8.

Table 1: *Mean square error* (MSE) on few-shot sinusoid regression.

|  | Sinusoid 5-shot | Sinusoid 10-shot |
|---|---|---|
| NP | 0.460 | 0.264 |
| MAML | 0.712 | 0.287 |
| BMAML | 0.409 | 0.200 |
| IPML(Ours) | **0.373** | **0.123** |

Table 2: Few-shot classification accuracy (%) on held-out Omniglot characters.

|  | Omniglot 1-shot 5-way | Omniglot 1-shot 20-way |
|---|---|---|
| NP | 95.9 | 55.3 |
| MAML | 98.7 | 92.5 |
| PN | 98.8 | **96.0** |
| IPML(Ours) | **98.8** | 94.0 |

Table 3: Few-shot classification accuracy (%) on mini-Imagenet test set.

|  | mini-ImageNet 1-shot 5-way | mini-ImageNet 5-shot 5-way |
|---|---|---|
| MAML | 48.6 | 65.9 |
| PN | 49.4 | **68.2** |
| PLATIPUS | 50.1 | - |
| BMAML | 49.1 | 64.2 |
| ABML | 45.0 | - |
| IPML(Ours) | **50.5** | 67.6 |

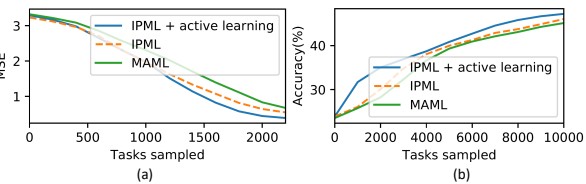

Figure 2: Results of active task selection on (a) 5-shot sinusoid and (b) 1-shot 5-way mini-ImageNet.

**Active task selection.** We can evaluate the effectiveness of the uncertainty measure arising from latent task posterior belief $p(\mathbf{z}|\mathbf{y}_{\mathcal{X}_t^s})$ by performing active task selection. Unlike previous works (Yoon et al., 2018; Finn et al., 2018) that can only perform active learning by querying data points, IPML can perform active learning by querying *tasks* and does not need the assumption of known task contexts in (Kaddour et al., 2020). In every iteration, a set of tasks are proposed with only the support set $(\mathcal{X}_t^s, \mathbf{y}_{\mathcal{X}_t^s})$ given; in image classification, it is usually one-shot. IPML will select among them the task with the maximum variance in $p(\mathbf{z}|\mathbf{y}_{\mathcal{X}_t^s})$ (with samples from the E step/SGHMC): $\arg\max_t \mathrm{Var}(\mathbf{z}|\mathbf{y}_{\mathcal{X}_t^s})$, and request for its query set to perform meta-training. This corresponds to a variance-based active task selection criterion. We test on both sinusoid regression and mini-ImageNet classification. Fig. 2 shows that the performance of IPML with active task selection improves over that of both MAML or IPML without active task selection, that is, it reaches a given MSE/accuracy with less training tasks. This shows that the uncertainty measure arising from $p(\mathbf{z}|\mathbf{y}_{\mathcal{X}_t^s})$ can be exploited to benefit meta-training.

**Measuring distance between tasks using latent task representation.** A most interesting question yet to be answered is the following: Does IPML learn a useful latent task representation? IPML learns to model the task through $\mathbf{z}$. If IPML learns the correct representation, then it can reflect patterns of task distribution in the latent space. While a solid criterion for assessing the correctness of learned latent task representation is hard to define, we can resort to an oracle (e.g., human expert with prior knowledge in designing the tasks). Our visualization of the latent task representation and quantitative evaluation of distance measure between tasks using *maximum mean discrepancy* (MMD) (Gretton et al., 2012) provide ways to assess the correctness of the learned task representation. We denote the set of samples from $p(\mathbf{z}|\mathbf{y}_{\mathcal{X}_t^s})$ as $\mathcal{Z}_t$. The MMD between tasks $t_1$ and $t_2$ can be calculated using

$$\mathrm{MMD}[\mathcal{H}, t_1, t_2] \triangleq \sup_{\varkappa \in \mathcal{H}} \left( |\mathcal{Z}_{t_1}|^{-1} \sum_{\mathbf{z} \in \mathcal{Z}_{t_1}} \varkappa(\mathbf{z}) - |\mathcal{Z}_{t_2}|^{-1} \sum_{\mathbf{z} \in \mathcal{Z}_{t_2}} \varkappa(\mathbf{z}) \right)$$

where $\mathcal{H}$ is a unit ball in the reproducing kernel Hilbert space with a radial basis function kernel.

We conduct experiments with the following 5-way 1-shot settings. **Setting A**: For subsampled Omniglot, we applied one rotation out of 4 possibilities $(0, \pi/2, \pi, 3\pi/2)$ uniformly across all the input images for each sampled task.[11] **Setting B**: For subsampled mini-ImageNet, a random artistic filter (normal, brighten, or darken) is applied for each sampled task. **Setting C**: For subsampled mini-ImageNet, a random artistic filter (3 different types of hue) is applied for each sampled task. **Setting D**: For subsampled mini-ImageNet, a random zooming (no zooming, zooming 3 times, or zooming 10 times) is applied for each sampled task. **Setting E**: On subsampled mini-ImageNet, a random artistic filter (normal, low contrast, or high contrast) is applied for each sampled task. **Setting A** has 4 types of tasks while **settings B to E** result in 3 types of tasks.

---

[11]In the previous experiment, the Omniglot dataset is augmented with rotations, but is random across the classes in a single task.

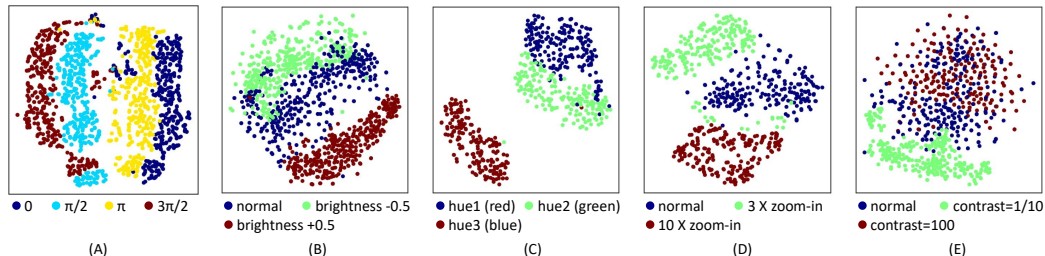

Figure 3: Visualization of latent task embeddings from settings A to E.

Table 4: Values of MMD metric between 4 types of tasks for Omniglot (setting A). Larger value means larger dissimilarity.

| Rotations | 0 | $\pi/2$ | $\pi$ | $3\pi/2$ |
|---|---|---|---|---|
| 0 | 0 | 1.166 | 0.594 | 1.134 |
| $\pi/2$ | 1.166 | 0 | 0.913 | 0.596 |
| $\pi$ | 0.594 | 0.913 | 0 | 0.917 |
| $3\pi/2$ | 1.134 | 0.596 | 0.917 | 0 |

Table 5: Results of meta-testing for training with real and generated tasks.

| Train on | Accuracy (%) |
|---|---|
| real | 73.83 |
| generated | 78.33 |
| real + generated | 88.16 |

For each setting mentioned above, we first train our models in IPML to converge, and then sample tasks from their latent task posterior beliefs (i.e., one sample of $\mathbf{z}$ per task). Finally, we visualize their latent task embeddings in the 2D space using TSNE (van der Maaten & Hinton, 2008). Furthermore, for setting A, we evaluate the distance measure between tasks using the well-known MMD metric with radial basis function kernels on the $\mathbf{z}$ samples. It can be observed from Fig. 3 and Table 4 that IPML successfully distinguishes 4 types of rotations for Omniglot. Both Fig. 3 and Table 4 contemporaneously show that flipping upside down (i.e., either right half of the embedding $0 \rightleftarrows \pi$ or left half of the embedding $\pi/2 \rightleftarrows 3\pi/2$) are reckoned to be closer tasks compared with rotation of $\pi/2$, thus revealing that our visualization and evaluation of distance measure between tasks are in accordance. From Fig. 3B to Fig. 3D, IPML successfully distinguishes different types of transformations on the tasks while revealing interesting facts: for example, tasks of high brightness are more isolated from that of low or normal brightness. Fig. 3E shows that tasks of low contrast are more distinct from that of normal or high contrast. The values of MMD metric for settings B to E and more details of the experiments are provided in Appendix A.6. On the overall, both the visualization and evaluation of distance measure between tasks reveal that IPML successfully learns useful latent task representations and even provides interesting insights.

**Synthetic task generation for Omniglot.** We assess the usefulness of latent task representation $\mathbf{z}$ by performing synthetic task generation. The training tasks we consider are three types of sub-sampled binary classifications: classification of characters A vs. B, B vs. C, and C vs. A, as in Fig. 4a. During meta-learning, we train a X-Net concurrently to learn to generate task-related input images (Sec. 3.3). The CVAE implementation of X-Net contains a decoder neural network with 3

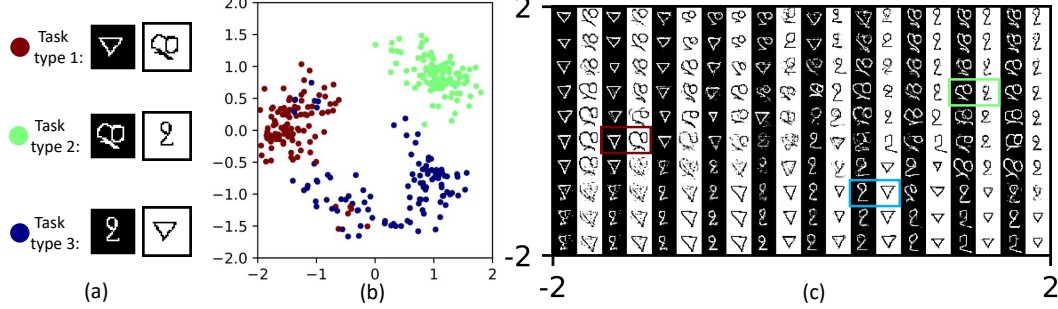

Figure 4: (a) TSNE visualization of (samples of) 3 types of binary classification tasks; images of black/white background are black/white samples ($y_{\mathbf{x}} = 1/y_{\mathbf{x}} = 0$). (b) Visualization of latent embedding of real tasks in (normalized) $\mathbf{z}$ space $[-2, 2]^2$. (c) Sampled generated task data by walking through the (normalized) $\mathbf{z}$ space $[-2, 2]^2$; note that the inversion of color is only for visualization to distinguish black and white samples. In training, NO images are inverted.

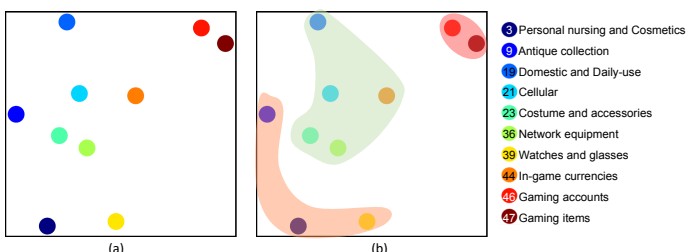

Figure 5: (a) TSNE visualization of latent task embedding of 10 meta-testing categories and (b) their analysis (see main text). Legend shows IDs and names of categories.

Table 6: Averaged meta-testing performance on 10 meta-testing categories.

|  | Accuracy (%) | F1 |
|---|---|---|
| IPML | **84.5** | **70.5** |
| Multi-task | 84.1 | 60.5 |

Table 7: Averaged meta-testing performance on 5 desired categories (IDs 19, 21, 23, 36, 44).

|  | Accuracy (%) | F1 |
|---|---|---|
| Setting A | **87.4** | **75.8** |
| Setting B | 86.4 | 74.4 |

hidden layers of size $[128, 128, 256]$ and ReLU nonlinearity, and a symmetric design of the encoder. After meta-training is completed, we continue to train the X-Net to converge. In this experiment, the dimension of $\mathbf{z}$ is set as 2, which further allows walking through such a latent space/embedding to visualize how the generated tasks map to their latent representations. Fig. 4b shows the latent embedding of real tasks. Fig. 4c shows the sampled synthetic tasks by walking through the latent space. It can be observed that X-Net successfully captures the task-dependent input distributions and can generate high-quality data of task type 1, 2, and 3 when sampled from their corresponding latent clusters (see samples of task type 1, 2, and 3 in the colored bounding boxes in Fig. 4c).

We further evaluate the quality of generated tasks by training on it. We hold out half of the images for each character during meta-training to construct the meta-testing tasks. The results are presented in Table 5. When training on both real and generated tasks, we first train on the generated tasks to converge and then train on the real tasks for another 30 iterations. It can be observed that compared to only using real tasks, a higher accuracy is achieved with training merely using generated tasks. When training on both real and generated tasks, a huge boost in accuracy is observed. We conjecture that due to their diversity, generated tasks (i.e., sometimes containing more ambiguous tasks) alleviate overfitting and provide a promising direction on meta-task augmentation.

**Real-world risk detection.** We perform experiments on a real-world risk detection dataset provided by an anonymous e-commerce company. The task is to classify whether an item in the online shop has risks (e.g., fraud, pornography, contraband). Such risks appear in different forms and in different categories (of items). It is hard to detect risks in different categories by training models separately for each category because some categories have only very limited amounts of black samples (i.e., $< 50$). The similarities of the detected risks in different categories, if discovered, can help improve the performance. Meta-learning is thus a suitable algorithm for its ability to perform (a) detection of risks across different categories of items and (b) adaptation to new categories. The input $\mathbf{x}$ of the dataset is the text (title and descriptions) embedding obtained from self-supervised learning, while its label is a binary variable indicating whether it contains risks (i.e., $y_{\mathbf{x}} = 1$ for black samples and $y_{\mathbf{x}} = 0$ for white samples). The data are separated by categories of items to yield 47 categories in total. Initially, we hold out 10 categories for meta-testing[12] while the rest are used for meta-training.

Table 6 shows results comparing the performance of IPML vs. a multi-task learning baseline.[13] It can be observed that IPML outperforms multi-task learning, which indicates its stronger ability to generalize to unseen categories. Fig. 5 visualizes the latent task embedding of the 10 meta-testing categories for analysis. IPML learns useful latent task representations: For example, from Fig. 5a, gaming-related categories with IDs 46 and 47 are mapped closely in the latent task space/embedding.

The individual meta-testing performance on the 10 meta-testing categories, which are given in Appendix A.3, can be further examined: For the five categories with IDs 19, 21, 23, 36, and 44 covered by the shaded light green zone in Fig. 5b, IPML outperforms multi-task learning by a large margin. They are mapped to the center of the latent task space (Fig. 5b), which may imply that IPML's adaptations to them can largely build on previous experiences of the meta-training categories and IPML's exploitation of such similarities allows their performance to improve over multi-task learning. For

---

[12]Their category names and IDs are given in Fig. 5.

[13]When testing on an unseen category, multi-task learning performs adaptation by randomly initializing its untied parameters for retraining on the few-shot support data.

the three categories with IDs 3, 9, and 39 covered by the shaded light orange zone, IPML does not have a performance advantage over multi-task learning. For the two categories with IDs 46 and 47 covered by the shaded light pink zone, both IPML and multi-task learning perform unsatisfactorily. As a matter of fact, for IPML, the categories with unsatisfactory performance (i.e., either covered by the shaded light orange or pink zone) are all mapped to be some distance away from the center, which indicates that they are likely considered by IPML as "outlier"/dissimilar tasks.

We further compare meta-learning on (A) the same setting as before by holding out the 10 meta-testing categories vs. (B) training on all categories in setting A as well as the dissimilar ones with IDs 3, 9, 39, 46, and 47. Table 7 shows results on the desired categories with IDs 19, 21, 23, 36, and 44. It can be observed that when a meta-learning model is trained to perform well (during meta-testing) on the desired categories/tasks, training alongside with dissimilar ones can compromise its performance. More details of the experimental settings and data preparation, experimental results, and analysis are given in Appendix A.3. We have also empirically compared the time efficiency of IPML against that of several meta-learning baselines and reported the results in Appendix A.7.

## 5 RELATED WORK

A number of meta-learning algorithms (Finn et al., 2018; Ravi & Beatson, 2018; Yoon et al., 2018) have proposed a Bayesian extension of the MAML framework (Finn et al., 2017). Their difference with IPML is that they model the uncertainty in the predictions with a set of particles (Yoon et al., 2018) or a variational distribution (Finn et al., 2018; Ravi & Beatson, 2018), which does not allow latent task modeling. The work of Rusu et al. (2019) introduces a generative model that decodes latent vectors into the meta-parameters, but does not scale well in the dimension of meta-parameters. In comparison, IPML explicitly represents each task as a latent continuous vector and models its probabilistic belief and is hence scalable in the dimension of meta-parameters. Moreover, MAML-based algorithms usually require evaluating computationally-intensive second-order derivatives of the meta-parameters during meta-training because they approximate the Bayesian inference through an inner loop of gradient descent. Although this issue can be addressed by methods such as first-order approximations (e.g., first-order MAML (Finn et al., 2017), Reptile (Nichol et al., 2018)) or implicit MAML (Rajeswaran et al., 2019) using implicit gradient, these works are not Bayesian. In contrast, our IPML algorithm naturally utilizes Bayes' rule to perform sampling during Bayesian inference and does not need second-order derivatives.

The work of Kaddour et al. (2020) uses latent information to perform active task selection, but assumes known task-descriptor (task context) which is usually unknown. The work of Garnelo et al. (2018) introduces the first use of stochastic processes (i.e., neural processes) in meta-learning and learns a heavily parameterized encoder to encode a dataset into its latent representation, which might introduce optimization difficulties and overfitting and can only output Gaussian posterior beliefs. In comparison, our IPML algorithm is the first to consider SGHMC in task adaptation/inference of meta-learning, which can capture a non-Gaussian posterior belief to achieve a better performance (Appendix A.4). Our IPML algorithm is also the first to explicitly model task-dependent input distributions, which is lacking in the literature. Such a modeling enables synthetic task generation of complex image classification tasks for the first time.

## 6 CONCLUSION

This paper describes a novel IPML algorithm that, in contrast to existing works, explicitly represents each task as a continuous latent vector and models its probabilistic belief within the highly expressive IP framework. Unlike existing works, IPML offers the benefits of being amenable to (a) the characterization of a principled distance measure between tasks using MMD, (b) active task selection without needing the assumption of known task contexts in (Kaddour et al., 2020), and (c) synthetic task generation of complicated image classifications via modeling of task-dependent input distributions using our X-Net. Empirical evaluation on benchmark datasets shows that IPML outperforms existing Bayesian meta-learning algorithms. We have also empirically demonstrated on an anonymous e-commerce company's real-world dataset that IPML outperforms the multi-task learning baseline and identifies "outlier"/dissimilar tasks which can degrade meta-testing performance.

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
