# OpenReview forum: "Meta-Learning with Implicit Processes"
_ICLR.cc/2021/Conference — Reject_

### Official Review · AnonReviewer2 · 2020-10-27
**We review the proposed method and the experiment results**

**Rating:** 5
**Confidence:** 3

**Review:**

This paper proposes an efficient meta-learning approach using implicit processes. Specifically, authors represent each task as a continuous latent vector and use expectation-maximization algorithm to perform meta-learning. The E step performs task adaption using stochastic gradient Hamiltonian Monte Carlo sampling method, while the M step optimizes meta-learning objective using these samples. Their framework can measure principled distance between tasks by maximum mean discrepancy (MMD) and generate synthetic tasks by task-dependent distribution. Finally, the authors validate their proposed framework on several benchmark datasets and real-world datasets. The novelty and originality of this paper is good by proposing new ideas and methods. In addition, the paper is well-organized and clearly written. We can quickly get to know what problem they are trying to solve, how they solve and what their results are.

Pros:
1.Representing each task as a continuous latent vector is very suitable when ones need to measure the task similarity, synthesize new tasks or actively select a task without the assumption of known task contexts.
2.The expectation-maximization algorithm based on the stochastic gradient Hamiltonian Monte Carlo sampling method can mitigate the enormous computation overhead by using first-order approximation without second-order derivative.
3.Authors demonstrate the effectiveness of this IPML on benchmark datasets and real-world tasks.

Cons:
1.The results of IPML in Table 2 do not outperform the baseline methods. I think authors could provide more analysis or reasons on results.
2.Authors should add more ablation study and analysis on why the proposed method would work. For example, how much is the contribution of representing each task as a continuous latent vector? How much is the contribution of the expectation-maximization algorithm?
3.Apart from the performance in the experiment, authors should also mention the efficiency of the IPML compared with other baseline methods. For example, the run time of the algorithm on computational expensive tasks.

---

> ### Author Response · Authors · 2020-11-24
> **Ablation study of effectiveness of IPML components and empirical evaluation of time efficiency of IPML are added, and clarifying your other concerns**
>
> We thank you for providing valuable suggestions and feedback, which we have seriously considered when revising our paper. We would like to address your comments and questions below:
>
> Regarding IPML sometimes not outperforming prototypical net (PN) in Tables 2 and 3, we have provided a reason in the second paragraph of Sec. 4 and will elaborate it further here: PN achieves a higher classification accuracy for $1$-shot $20$-way Omniglot and $5$-shot $5$-way mini-ImageNet, likely due to the use of extra classes during training (Snell et al., 2017). Specifically, though meta-testing involves $N$-way classification for all tested algorithms, the training of PN requires more than $N$ classes, that is, $60$-way classification which is also the setting adopted in (Snell et al., 2017). As a result, since PN utilizes more information from the extra classes during training, it is reasonable to expect that PN achieves a higher classification accuracy at times. We have included the above analysis in our revised submission.
>
> About adding more ablation study and analysis on why our proposed IPML would work well, we have included such results in Appendix A.4 of our revised submission, which are now reorganized to ease understanding. Specifically, in Appendix A.4.1, we have empirically shown a general performance decrease if SGHMC in our IPML algorithm (Sec. 3.1) is replaced by variational inference (VI), hence revealing the effectiveness of using SGHMC over VI. In Appendix A.4.2, we have empirically shown a significant performance improvement of our EM algorithm for IPML (Sec. 3.1) over the variational Gaussian process (GP) framework proposed by Ma et al. (2019), hence reflecting the effectiveness of our EM algorithm over the variational GP framework. In Appendix A.4.3, we have empirically shown a substantial performance improvement by using the continuous-valued latent task vector as a mask (Sec. 3.2) rather than in a naive concatenation, hence demonstrating the effectiveness of our delicate design of the coupling of the latent task vector with the generator over the naive design.
>
> As you have suggested, we have empirically compared the time efficiency of IPML against that of several baseline methods on the anonymous real-world e-commerce company's risk detection dataset and reported the results in Appendix A.7 of our revised submission; we have referred to this empirical comparison from Sec. 4. To summarize here, the results show that the time efficiency of our IPML algorithm ($1.27$s/iteration) is comparable to that of the first-order MAML baseline ($1.01$s/iteration) since (a) the forward passes (E step/SGHMC) are not that slow compared with the inner optimization loops of first-order MAML, and (b) the cost of backward passes of IPML is similar to that of first-order MAML. On the other hand, the BMAML baseline is not as time-efficient ($3.74$s/iteration) due to the use of particles.
>
> We sincerely hope that our clarifications above will improve your opinion of our work.

---

### Official Review · AnonReviewer4 · 2020-10-27
**Solid paper - Though, better comparison to Neural Processes is needed**

**Rating:** 6
**Confidence:** 4

**Review:**

=== Summary ===

The paper proposes a meta-learning method based on implicit process (IP) framework in which each task is represented by a latent vector. The IP setup for meta-learning seems identical to that of Neural processes [1]. In that, the key challenge for adaptation to a task based on a context/support set $(X_c, Y_c)$ is inferring the distribution $p(z|X_c, Y_c)$ over latent vectors. While [1] use amortized (variational) inference with a variational Gaussian distribution to approximate the true $p(z|X_c, Y_c)$, the paper proposes to use stochastic gradient Hamiltonian Monte Carlo (SG-HMC) for sampling latent state vectors from $p(z|X_c, Y_c)$.

In the experiment section, the paper compares the proposed method (IMPL) against various previous methods, mostly gradient-based meta-learners, on common benchmarks, demonstrating competitive results. In addition, the paper provides a detailed inspection of the latent space and argues that IMPL could be used for active task selection and synthetic task generation.

=== Main argument ===

The IP meta-learning setup of the paper is well-known in the meta-learning community (e.g. [1]) and thus not a contribution. Unlike to previous work (e.g. [1]) which uses the IP framework for meta-learning, the paper proposes to use SG-HMC instead of amortized (variational) inference for approximating p(z|X_c, Y_c), resulting in a new, sound meta-learning algorithm. This can be considered the methodological main contribution of the paper.

The experiment section features standard, well-accepted meta-learning benchmarks and demonstrates that IMPL is not only theoretically sound but also performs well in practice. The analysis and visualization of the learned latent task representation is interesting and provides nice qualitative evidence that proximity of task representation vectors in the latent space reflects the human-perceived task similarity to a fair degree.

Unfortunately, the paper’s experiment section does not address some of the core questions: How does IMPL compare to Neural Processes (NPs)? What are the pros and cons of replacing the amortized (variational) inference with SG-HMC? E.g. How much performance increase does the arguably much more complex and computationally demanding approximate inference method buy us? Since, the key contribution of the paper is replacing the approximate inference method of NPs with SG-HMC, an empirical in-detail comparison with NPs would strengthen the paper a lot!

In the related work section, the paper argues that a limitation of NPs is the Gaussian variational distribution in the latent space. However, since the latent space is meta-learned and thus quite arbitrary - this is not a very convincing argument. Empirical evidence for the hypothesized shortcomings of NPs would be much better.

Furthermore, it remains unclear why/how active task selection could be relevant in any real-world setting. Motivating the setup with a real-world use case would strengthen this aspect of the paper. The real-world risk detection experimental results in the appendix are nice and interesting. Putting more of the results in the main part (there is still some space) and briefly discussing them could give a nice argument for the benefits of the latent task representation inherent in IMPL - i.e. it provides a straightforward way for identifying tasks that are very dissimilar from the rest - excluding such outlier tasks may increase performance.

=== Overall assessment ===

In the current form, I see the paper slightly above the acceptance threshold. The proposed meta-learning algorithm based on IPs is sound and the extensive experiments demonstrate that it works well in practice. However, the only innovation of IMPL over Neural Processes (NPs) is replacing variational inference by SG-HMC. Thus, the algorithmic contribution is limited. What the advantage of IMPL over NPs is, has not been addressed sufficiently. If a proper experimental comparison to NPs is added to the paper, I am happy to increase my score.

[1] Garnelo, Marta, et al. "Conditional neural processes." arXiv preprint arXiv:1807.01613 (2018).

---

> ### Author Response · Authors · 2020-11-24
> **Comparison to Neural Processes is added, and clarifying your other concerns**
>
> We thank you for providing insightful comments and advice, which have been incorporated into our revised submission.
> We would like to address your comments and questions below:
>
> We would like to suggest that our contribution is not limited to replacing the amortized variational inference (VI) in NP to SGHMC, but also in designing the coupling of the latent task vector with the generator to achieve competitive meta-learning performance (Sec 3.2 and Appendix A.4.3 of our revised submission) and exploiting X-Net for synthetic task generation (Sec 3.3).
>
> We would like to address your main concern by including experimental results of NP (Tables 1 and 2) and an analysis to explain the performance difference between IPML and NP (Sec. 4) in our revised submission. For both sinusoid regression (Table 1) and Omniglot (Table 2), NP performs unsatisfactorily as compared to IPML, likely because (i) it performs amortized VI of $\mathbf{z}$ through a heavily parameterized encoder which may introduce optimization difficulties and overfitting, and (b) the encoder of NP takes in the simple concatenation of $(\mathbf{x},y_{\mathbf{x}})$ and thus does not explicitly capture the $\mathbf{x}\rightarrow y_{\mathbf{x}}$ relationship in the support set. An ablation study of such limitations is presented in Appendix A.8 of our revised submission.
>
> We agree that since the latent space is meta-learned and thus quite arbitrary, the Gaussian variational distribution may not be a major limitation. However, we think that the Gaussian variational distribution can still be a (minor) limitation of NP, which is supported by the empirical evidence in Appendix A.4.1 of our revised submission: By replacing SGHMC in our IPML algorithm with VI, a performance decrease is observed in 4 of the 5 test cases. This can only likely be explained by the limited expressiveness of the Gaussian variational distribution in representing the latent task posterior belief.
>
> We have also empirically compared the time efficiency of IPML vs. NP (and other baseline methods) on the anonymous real-world e-commerce company's risk detection dataset and reported the results in Appendix A.7 of our revised submission. Due to the use of amortized VI, NP incurs a slightly shorter time in finetuning/adaptation to new tasks. However, since NP introduces a heavily parameterized encoder, its model backward pass/update incurs more time.
>
> Regarding how active task selection could be relevant in the real-world setting, we would like to motivate it with a real-world use case: In our real-world risk detection experiment (Sec. 4), a considerable fraction of black and white labels (i.e., risk or no risk) are labeled manually due to the rapidly evolving nature of risks or fast emergence of new variants of risks. However, for the anonymous e-commerce company, the labeling budget is limited (e.g., a human can only label a fixed amount of items per day). This will require us to select the best sequence of tasks to be labeled to best improve the model. Such problem is crucial and is essentially an active task selection problem that can be tackled using our proposed IPML.
> We have included the above discussion in Appendix A.2.3 and referred to it from Sec. 1 of our revised submission.
>
> We agree with you that the real-world risk detection experiment is interesting and have included it in Sec. 4 in the main paper as you have suggested.
>
> We earnestly hope that our clarifications above will improve your opinion of our work.

---

### Official Review · AnonReviewer1 · 2020-11-01
**A solid paper but unclear motivation**

**Rating:** 6
**Confidence:** 4

**Review:**

This paper proposes Implicit Process Meta-Learning (IPML) where each task is represented as a continuous latent vector $\mathbf{z}$, and corresponding data points are described as function values evaluated at an implicit process conditioned on the task latent vector $\mathbf{z}$. To conduct the intractable inference, a stochastic gradient Hamiltonian Monte Carlo (SGHMC) algorithm is employed. A VAE-like network called X-Net is trained simultaneously to generate synthetic tasks from the task latent vectors. The experimental results demonstrate that the proposed algorithm shows decent performances on few-shot classification tasks, and the task latent vectors indeed represent a meaningful space of the tasks on which measuring distances between tasks and detecting outlier tasks are possible.

Overall I find this paper is clearly written. Various experiments are conducted to demonstrate the benefits of the proposed model. Personally, I like the implicit process framework, and I guess this is a nice application of the idea to meta-learning.

Since the performance of the proposed algorithm is not overwhelming, I presume the main advantage of the proposed algorithm is its ability to model a proper latent space of tasks.  My first concern is that whether this is a property only for the proposed IPML. For instance, consider a prototypical net. Although not being as principled as IPML, a prototypical net can also represent a task via prototypical vectors and consider them as a latent representation of the task (e.g., just take the average of all prototype vectors to get a single representative of a task).  Is this space completely meaningless so measuring the distance metric doesn't reflect the semantic relationship between tasks at all?

Also, as far as I understood, the ability to construct a meaningful latent space of tasks is not solely from the implicit process itself but aided by the additional X-Net learning a generative model over the data points in the tasks. What if we conduct measuring the MMD between tasks for the model without X-Net (not between the generated tasks and an existing task, but between existing tasks semantically apart)? Moreover, how the prototypical-like model with a generative model attached compare to the proposed one on synthetic task generation? The reason why I'm concerned about this is training a generative model and generating an image from it is usually quite expensive. For instance, one can easily imagine that training X-Net for miniImageNet or tiredImageNet and generating images from it might be effective.

Some minor questions
- Figure 3 shows the shift in the latent space when tasks data are manipulated. But is it really demonstrate the semantic difference between tasks? I guess a representation from an ordinary classifier might also show this because the input "images" were shifted using non-trivial operations. I guess this point may be made clear by having a look at the representations of semantically different tasks (e.g., classification of different classes) without being modified at the image level.
- For a binary classification task (Figure 4), when generating a task from a task vector $\mathbf{z}$,  as far as I understood, each data $\mathbf{x}$ is generated conditioned on $\mathbf{z}$ and then corresponding label $y$ is generated. Are the generated data is well balanced between positive and negative classes?
- Have you considered using SGLD instead of SGHMC? If so, did you observe any empirical degrade in the performance?

---

> ### Author Response · Authors · 2020-11-24
> **Clarifying the motivation of our work, among others**
>
> We thank you for appreciating our contributions and providing valuable feedback, which have been taken into account when revising our paper.
>
> Indeed, a main advantage of IPML is its ability to model a proper latent space of tasks. In contrast to the other works that model a task as a discrete cluster or a single latent vector, we model a task as a probabilistic belief in the latent space. The three distinct benefits with doing so are that IPML becomes amenable to active task selection without needing the assumption of known task contexts, the characterization of a principled distance measure between tasks using maximum mean discrepancy (MMD), and synthetic task generation by modeling task-dependent input distributions.
>
> As you have mentioned, a prototypical net may produce a latent vector of the task by taking an average of all prototypical vectors. However, firstly, such an averaging is not principled. As a result, a task may lose its semantic information and the L2 distance metric may not reflect a meaningful semantic relationship between tasks. For example, the averaging of prototypical vectors of each class cannot be distinguished from a new classification task whose class labels are just the permutation of the original ones. Secondly, while the prototypical net can only handle classification tasks, our IPML algorithm can cater to both classification and regression tasks.
>
> We would like to clarify that the ability to construct a meaningful latent space of tasks is in fact solely learned from the implicit process during meta-training and does not need the help of X-Net. The X-Net just uses such a latent representation (as contextual information) to additionally model task-dependent input distributions. In our work, we train X-Net concurrently with IPML. We can also train X-Net after meta-learning, which we have tried and a similar outcome results. As for measuring the MMD between existing tasks semantically apart, Fig. 3 shows existing (i.e., not generated) tasks being semantically apart, while Table 4 (Sec. 4) and Table 15 (Appendix A.6) show the corresponding MMD values between these existing tasks. These results show that IPML can capture the difference/distance between tasks via MMD. About your concern that training a X-Net generator from scratch is time-consuming, we think our IPML can also use a pretrained generator and finetune it by feeding it with additional contextual information ($\mathbf{z}$), which we will investigate in our future work. We think that finetuning the generator is necessary so that it can (a) adapt to the domain of the current task inputs and (b) generate task-dependent inputs.
>
> Regarding your first minor question that the shift in the latent space may not demonstrate the semantic difference between tasks, we have additionally provided a visualization of latent input embeddings based on an ordinary classifier in Appendix A.6 (Fig. 6) of our revised submission. A comparison of Fig. 6 vs. Fig. 3 shows that when tasks are modified at the image level, an ordinary classifier may be able to distinguish the differences in hue (i.e., setting C), but cannot distinguish well the differences in rotation, brightness, zooming, and contrast. This comparison gives evidence showing that the ordinary classifier cannot capture the semantic difference between tasks while our proposed IPML algorithm can learn such semantic difference between tasks. The above discussion is included in Appendix A.6 of our revised submission.
>
> About your second minor question on whether the generated binary labels are well balanced, for our synthetic task generation experiment in Sec. 4, we have observed that the ratio between labels of generated images from our IPML is quite balanced and we did not observe mode collapse of our CVAE-based X-Net. Nevertheless, we have provided a method in Appendix A.5 that guarantees the generation of balanced labels using our proposed doubly-contextual X-Net, which we have referred to from Sec. 3.3 of our revised submission.
>
> For your last minor question about SGLD, we have not tried SGLD yet. We will implement other sampling methods to investigate the difference in the performance in our future work.
>
> We hope that our clarifications above will improve your opinion of our work.

---

### Decision · Program_Chairs · 2021-01-07
**Final Decision**

**Decision:**

Reject

**Comment:**

This paper sits at the borderline: the reviewers agree it is a well-written and interesting paper, but have concerns about efficiency as well as a comparison with the neural process (the authors did include a revision with this comparison, though the numbers they report are worse than in the original neural processes paper on the same experiment). Ultimately, this paper probably requires another round of reviews before it is ready for publication.